# The Processivity of Telomerase: Insights from Kinetic Simulations and Analyses

**DOI:** 10.3390/molecules26247532

**Published:** 2021-12-13

**Authors:** Clive R. Bagshaw, Jendrik Hentschel, Michael D. Stone

**Affiliations:** 1Department of Chemistry and Biochemistry, University of California at Santa Cruz, Santa Cruz, CA 95064, USA; jhentsch@ucsc.edu; 2Element Biosciences, 9880 Campus Point Drive, San Diego, CA 92121, USA

**Keywords:** reverse transcriptase, polymerase, numerical analysis, product distributions

## Abstract

Telomerases are moderately processive reverse transcriptases that use an integral RNA template to extend the 3′ end of linear chromosomes. Processivity values, defined as the probability of extension rather than dissociation, range from about 0.7 to 0.99 at each step. Consequently, an average of tens to hundreds of nucleotides are incorporated before the single-stranded sDNA product dissociates. The RNA template includes a six nucleotide repeat, which must be reset in the active site via a series of translocation steps. Nucleotide addition associated with a translocation event shows a lower processivity (repeat addition processivity, RAP) than that at other positions (nucleotide addition processivity, NAP), giving rise to a characteristic strong band every 6th position when the product DNA is analyzed by gel electrophoresis. Here, we simulate basic reaction mechanisms and analyze the product concentrations using several standard procedures to show how the latter can give rise to systematic errors in the processivity estimate. Complete kinetic analysis of the time course of DNA product concentrations following a chase with excess unlabeled DNA primer (i.e., a pulse-chase experiment) provides the most rigorous approach. This analysis reveals that the higher product concentrations associated with RAP arise from a stalling of nucleotide incorporation reaction during translocation rather than an increased rate constant for the dissociation of DNA from the telomerase.

## 1. Introduction

Nucleic acid polymerases catalyze the elongation of a DNA or RNA strand, (N)_n_ by the addition of one nucleotide at a time, derived from a nucleoside triphosphate substrate, according to Figure 1:

The correct nucleotide for incorporation is selected according to the sequence of a complementary template strand. Typically, after the addition of a nucleotide, the newly formed duplex product moves out of the active site to make way for next free base in the template sequence. However, the duplex usually remains attached to the polymerase during this translocation. Accordingly, most polymerases are highly processive and many hundreds or thousands of bases may be incorporated before the product nucleic acid dissociates from the polymerase.

The processivity of nucleic acid polymerase activity has been defined in several ways. At each step in the polymerase reaction, the enzyme may either continue to catalyze the elongation reaction or dissociate [1]. The relative probability of these events, p, is defined as the microscopic processivity [2,3,4,5], and is calculated from the rate constants *k_fi_*/(*k_fi_* + *k_di_*) for the *i*th step the reaction Figure 2:

Here, we focus on telomerases, which are reverse transcriptases that use a short integral RNA template to elongate the ends of linear chromosomes [6]. Human telomerase incorporates an average of a few tens to a hundred nucleotides [7], before dissociating, in contrast to the thousands incorporated by many DNA polymerases. In human telomerase, the RNA template includes six nucleotides that direct the sequential incorporation of the complementary nucleotides, GGTTAG, in a moderately processive manner. This processivity is defined as the Nucleotide Addition Processivity (NAP) and corresponds to single nucleotide addition steps. When this phase is complete, the template dissociates from the newly synthesized DNA and repositions itself in the active site and the DNA rebinds to the RNA template to initiate another round of nucleotide addition (Figure 1a). These steps are collectively referred to as translocation. The result of this process is revealed by gel electrophoresis, which shows an enhanced intensity every sixth band, indicative of a lower processivity associated with the translocation steps (Figure 1b). We denote these intense *repeat addition* bands, RA-bands to distinguish them from the intervening NA-bands which correspond to single *nucleotide additions*. In the analysis of such gels, often only the RA-bands are considered and the processivity is reported in terms of progression between the intense bands: the so-called Repeat Addition Processivity (RAP). The existence of RAP indicates that the DNA maintains contact with the telomerase when it transiently dissociates from the RNA template via protein-DNA interactions at so-called anchor sites (Figure 1a) [8,9,10]. However, the anchor site interactions must also be dynamic to allow the growing DNA strand to elongate. An alternative explanation for RAP is that the DNA does not fully dissociate from the RNA but loops back to reset the DNA/RNA hybrid ready for further extension [11]. The increased band intensity associated with the translocation could arise either because the intrinsic probability of DNA dissociation is higher than that during the NA-steps, or the forward progression is slower, allowing more time for the product to dissociate (or result from a combination of both events). Only kinetic measurements can resolve this question. Depending on the model under consideration, the transition A to B and so forth in Figure 2 may represent the addition of one nucleotide in the context of NAP, or the addition of a hexanucleotide repeat when the scheme is used to model only the telomerase RAP process. In the latter case, the NA-bands are often omitted from the analysis for simplicity because they are relatively weak in intensity (typically 10–20%) compared with the RA-bands [7,12].

If the microscopic processivity value is the same at each step in the reaction (i.e., p equals P, the macroscopic processivity [13]), then the concentration of each released product plotted against the step number will follow an exponential decay curve, *when the system approaches its end-point distribution* [4,6]. The average or macroscopic processivity may also be described in terms of the median product length (= −ln2/decay constant = R_1/2_; [7]) or the mean product length (R_mean_ = 1/decay constant; [4]). R_1/2_ and R_mean_ are analogous to the half-time and time constant, respectively, of a first-order reaction. In practice, it is unlikely that the processivity will be identical at each step and this will be revealed in deviations from an exponential function [14]. Processivity has also been defined in an empirical way using a processivity index [15], which defines the fraction of the products that exceed a certain product length. While a single macroscopic processivity value or index is useful for comparative purposes for determining the effect of conditions, regulators or mutations on processivity, microscopic processivity values are required for a mechanistic understanding of the effect of local sequence and structure on the catalytic activity of telomerases.

Recently, we found that, under some conditions, the band intensities in a primer extension assay of telomerase were modulated in a “pattern of four”, indicative of G-quadruplex formation affecting the microscopic processivity values of the RAP steps [12]. Microscopic processivity values were determined from kinetic analysis of time courses using non-linear least squares regression (DynaFit [16,17]) for the model shown in Figure 2. Here, we present the details behind this procedure and test the DynaFit program using simulated data to check for factors that can affect the analysis. In addition, these simulated data allowed a quantitative assessment of systematic errors that can arise in other established methods for determining processivity values. These errors can arise from (i) analysis of an incomplete data set due to the limited resolution of high molecular weight products in gel-based assays; (ii) analysis at an incubation time before a stable (i.e., near equilibrium) product distribution has been established; (iii) omitting the weak NA-bands from the analysis; (iv) failure to maintain “single hit” conditions so that dissociated products rebind to the enzyme and undergo further elongation; and/or (v) analysis of bands, which comprise both bound and free products. While some of these problems can be addressed empirically (e.g., changing the incubation time until the banding pattern becomes effectively constant), the use of simulated data is instructive to gauge the magnitude of these errors and to help in experimental design.

## 2. Results

### 2.1. Primer Extension Assays

Before discussing the analysis of simulated data, it is useful to review the nature of the experimental approaches and their variations. A typical primer extension assay involves adding telomerase to the primer DNA (e.g., the 18-mer (TTAGGG)_3_), followed by addition of the nucleotide substrates (dATP, dGTP and dTTP in the case of human telomerase). After an appropriate incubation time, the reaction is stopped by the addition of EDTA and SDS, phenol-chloroform extracted and the DNA precipitated with ethanol, prior to separation by gel electrophoresis [6,7,12]. If the DNA primer is radiolabeled, then the radioactivity in each band, whose molecular weight is increased by one nucleotide addition, is directly proportional to its concentration (Figure 1b). Alternatively, one of the deoxynucleotides may be radiolabeled, which has the advantage that more counts will appear in the higher molecular weight bands. This compensates for their lower concentration at the extremity of the DNA product exponential distribution. In this case, the product concentration in terms of DNA molecules needs to be adjusted according to the number of labeled nucleotides in the extended sequence. While advantageous in terms of signal strength, this approach may limit the molar concentration of the labeled nucleotide used in the assay to ensure a reasonable fraction of the counts are incorporated. Furthermore, this procedure complicates the assessment of the contribution from the unresolved bands, which is necessary for some analytical procedures discussed below. One problem in both assays is the potential for dissociated products to rebind and undergo further elongation. This can be minimized by restricting the analysis to early time points, such that the labeled primer remains in large excess. A better solution is to add a large excess of unlabeled primer soon after the assay is initiated (a pulse-chase assay) to limit the rebinding of all labeled species.

Following electrophoresis, the gel is typically analyzed using a phosphorimager. In order to retrieve data of sufficient quality to warrant quantitative analysis, a number of procedures are incorporated to minimize random and systematic errors. During phenol/chloroform extraction, the volume of the aqueous layer is increased so that any denatured protein at the interface is not carried over to the gel where it might interfere with DNA migration. With this precaution, the number of counts remaining in the well is < 0.5% that of the resolved bands and there is no evidence of smearing of intensity beneath the wells. Variations arising from pipetting errors and extraction efficiencies, which typically span about 10%, can be accommodated by normalizing the total counts of each lane. The total counts should also be examined to make sure there is no systematic error across lanes that might reflect differential extraction of low and high molecular weight DNA products. The relative product concentrations are determined from the counts in each band and, in some cases, are calibrated absolutely using the labeled primer intensity as a standard (see below).

The intensity and resolution of the bands become progressively lower with increasing molecular weight. Telomerases present a further challenge in that the 5 NA-bands between each pair of RA-bands are an order of magnitude weaker in intensity and often ignored in the analysis. NA-bands are poorly resolved beyond about the 5th RA-band, while the RA-bands themselves may be analyzed up to around 20 to 25 repeats (equivalent to 120 to 150 total added nucleotides: Figure 1b and Appendix A). Analysis of gel band intensity can be performed with various degrees of sophistication from manual definition of peak boundaries using a tool such as the Analyze Gel option in ImageJ (https://imagej.nih.gov/ij/ accessed on 7 December 2021) to least-squares fitting to a series of Lorentzian line shapes [18,19,20]. Ultimately, the analysis is limited by insufficient resolution of the high molecular weight bands and uncertainty in the baseline counts. However, in some procedures it is important to estimate the net contribution from the unresolved bands.

### 2.2. Analysis of Processivity from the “End-Point” Product Distribution

We start by considering a simplified mechanism, where the microscopic processivity is the same at each step of the reaction (p = P), in order to explore the systematic distortions of the data that arise during various analytical procedures. (Table 1).

Figure 2 can be used to analyze RA- to RA-band transitions, where the NA-steps are ignored. With constant *k_f_* and *k_d_* values, the RA-product concentrations show a discrete exponential distribution as the reaction approaches equilibrium. For example, with P = 0.9 for RAP, 90% of the product undergoes further extension in units of the hexanucleotide repeat while 10% dissociates at each RA-step (Table 2). Consequently, after seven repeats, the product is reduced in concentration to 0.9^7^ = 0.478, which is close to the median product length expressed in terms of number of repeats until the [product] is reduced to 50% (R_1/2_ = 0.69).

Fitting a continuous exponential decay function to the product distribution provides a direct method for defining the processivity (Figure 2a: decay constant = 0.105 per repeat). From this decay constant, R_1/2_ equals Ln(2)/decay constant and the macroscopic processivity, P equals e^(−decay constant)^. Variations of this method involve taking the logarithm of the product concentration which gives a linear plot, but this can distort the analysis. Linear transformation prior to data fitting biases the result due to distortion of the error distribution [21,22]. In addition, it weights the fit towards higher repeat numbers where the relative experimental error is larger because of low intensity and poor resolution from neighboring bands. 

In the case of the von Hippel plot, −Log(I_n_/I_total_) versus repeat number [4], the intensity of a band at position n (I_n_) is normalized with a constant denominator, I_total_, the total intensity. Note that von Hippel et al. applied their equation to DNA polymerases and not telomerases, but the analysis is included here to compare with the logarithmic plot of Latrick and Cech [7] which has different characteristics. In the von Hippel plot, errors in the total intensity due to truncation of the analyzed gel area (Figure 2b) cause a small shift in the intercept value, but not the slope. The latter therefore potentially yields an accurate processivity estimation (apart from the error weighting problem of log plots referred to above), as confirmed in Figure 2b. Note that in the original article [4], processivities were given in terms of the mean number of repeats, R_mean_ (=1/decay constant; Figure 2a) rather than R_1/2_.

In the Latrick and Cech plot [7], the data are transformed by normalizing to the Fraction Left Behind (FLB: Table 1). This plot is based on a cumulative normalized exponential function which smooths out local variations in the microscopic processivity values. While this appears to assist in determining a macroscopic processivity value, P, it can also introduce systematic error. Any underestimate in the total concentration due to analyzing a finite number of bands causes the plot to curve at high repeat numbers and to under-estimate the processivity (Figure 2c). In the case of experiments using a labeled primer, the error is readily corrected by adding the contribution from the unresolved peaks in the normalization process. This contribution can be estimated from the relative area (pixel count) of the unresolved peaks in the gel scan compared with the resolved peaks which are included in the analysis (Appendix A).

The analysis of microscopic processivity by the method of Peng et al. [5], also involves normalization to a running accumulated product intensity. This plot likewise suffers from curvature if only a limited number of bands are included in the normalization (Figure 2d). However, the correction for the total product concentration returns the input P = p = 0.9 at each step of the reaction.

### 2.3. Errors due to Incomplete Equilibration

An exponential distribution of product concentrations arises only when the reaction nears its “end-point” value. This refers to the equilibrium concentration for a pulse-chase experiment where all the labeled products dissociate and are prevented from further extension by the large excess of unlabeled primer. Experimentally, the time to reach a near end-point distribution is determined empirically from the incubation period where no further change in the resolvable product band intensities is detected. This time may be incorporated into a standard protocol but its validity is not always checked for different experimental conditions or number of bands analyzed. To determine the effect of incomplete equilibration on the analysis of processivity, a reaction was simulated according to Figure 2 with k_f_ = 0.18 min^−1^ and k_d_ = 0.02 min^−1^ at each step in the reaction, corresponding to a processivity of 0.9. The values for the rate constants were chosen to roughly match the time course of product formation in telomerase assays under the conditions used by Jansson et al. [12]. In this simulation, 5 nM telomerase was mixed with 50 nM labeled primer DNA and, after a short time interval (5 min in the simulation analyzed in Figure 3), 10 μM cold primer was added.

It can be seen that each product reached its peak concentration at intervals of about 5.5 min (Figure 3a), controlled by its mean lifetime (=1/*k_f_*). For subsequent analysis, samples were analyzed every 10 min for a total incubation time of 200 min (Figure 3a), in line with a typical experimental protocol. After about 120 min the products corresponding to the first 15 repeats had fully dissociated and showed an exponential distribution with respect to repeat number (Figure 3b). The decay constant of 0.105 repeat^−1^ corresponds to the input processivity of 0.9 and matches that of the equilibrium analysis in Figure 2a. However, after 90 min incubation (85 min after chase), the total product distribution deviates from a single exponential function from about the 9th repeat (RA-band J) onwards (Figure 3b, open circles). The curve falls above the exponential function because some product remains bound to the telomerase, so increasing the total product concentration for those species. In some experimental protocols [23], the incubation mix is subject to a separation step before gel electrophoresis, so that the bands represent the free rather than total product. This procedure reduced the deviation, but now the data fell below exponential function from about the 11th repeat (Figure 3b solid circles). This simulation indicates that the best approach is to wait for the reaction (>120 min for 15 RA-bands) to reach a stationary end-point, after which the total product equals free product and avoids the need for a separation step. The latter may itself cause problems if the bound product dissociates on the time scale of the separation procedure.

Analyzing the same data using the Latrick and Cech procedure [7] for the first 15 repeats, with correction for the truncation error discussed above, gave a linear plot for the product distribution at 200 min and a derived P = 0.89, close to the input value (Figure 3c). However, the distribution at 90 min gave a non-linear plot, although the initial slopes for the total and free product yielded P = 0.87 and 0.91, respectively, which are reasonably close to the input P value. Note that the deviation from linearity is seen by the 2nd repeat because the smoothing effect of a cumulative plot mixes “good” data (bands B to H) with the “bad” (bands J to P).

Peng plots [5] yielded the correct microscopic processivity values for the 200 min data, but progressively deviated from about the ninth repeat for the 90 min data (Figure 3d). In particular, the total product underestimates the microscopic processivity. This is a consequence of the contribution from the bound product, most of which would continue to elongate rather than dissociate. On the other hand, the free products at 90 min overestimated the processivity beyond the 11th repeat because the dissociated products have not risen to their final values.

Finally, an attempt was made to fit the complete time series at 10 min sampling intervals by global fitting using numerical methods [12,16,17] to determine *k_f_* and *k_d_* at each step of the reaction for the first 15 repeats. When the dataset included data up to 200 min, the estimated *k_f_* and *k_d_* values were accurate and defined the microscopic processivity, *k_f_*/(*k_f_* + *k_d_*) = 0.90. When the data only extended to 90 min, the estimated processivity deviated from about repeat 11. Note the slight improvement in the microscopic processivity estimates for the 90 min data compared with Peng analysis (Figure 3d) for the total product intensity. This arises because the kinetic information allows an estimate of the partitioning of the total product between free and bound states between the ninth and 11th repeat but, beyond the 11th repeat, insufficient dissociation occurs to allow an accurate estimate of the *k_d_* value. In practice, it is better to solve this problem by obtaining data at longer incubation times or restricting rigorous analysis to low repeat numbers, where the products have fully dissociated. We show below that the deviations, which arise from ignoring or incompletely resolving the NA-bands can give rise to systematic errors in the fit, and therefore it is better to determine end-point concentrations experimentally rather than relying on the extrapolation of a kinetic model.

### 2.4. Errors Arising from Analysis of Steady-State Distributions

The time course of primer extension assays, in the absence of a chase with excess unlabeled primer, give more complex profiles (Figure 4). Each product showed a lag phase depending on the repeat number, a burst phase dominated by bound product, followed by a near-linear steady-state phase as the free product accumulated.

An experimental protocol invariably requires a compromise in selecting an optimal incubation time. Long incubations are required for the later products to appear during which time the early products may rebind and invalidate the assumption of “single hit conditions” required for an accurate assessment of the processivity. A rule of thumb is sometimes used that the primer should not be depleted by more than 10% during the assay. Analysis of simulated profiles confirms these complications. Under the same conditions as used for the simulation in Figure 3a, but without the chase, the reaction barely reached a steady-state as evident from the curvature of the final phase of early products (Figure 4a). This arises because of the relatively high ratio of telomerase (5 nM) to labelled primer (50 nM), which caused depletion of the primer to 68% of the starting concentration by 150 min. The curvature arose from products rebinding in competition with the primer. In the simulation, the rate constant for product rebinding was the same as initial primer binding (0.1 nM^−1^ min^−1^) and therefore degree of competition was proportional to the relative concentration of each species. While the product distribution at 200 min follows an exponential function, the extracted processivity, P = 0.91 (R_1/2_ = 7.5 repeats) slightly overestimates the processivity. Kinetic analysis using DynaFit, assuming a similar association rate constants for all species, provided a better estimate of the true processivity (P = 0.890 to 0.904) because it modeled the contribution from product rebinding. In practice, the association rate constants for each product are unknown, thus limiting the value of this approach

Reducing the telomerase ten-fold to 0.5 nM overcomes the multiple hit problem, so that the free product is formed linearly (Figure 4b). Here, the primer concentration was 96% of the starting value at 150 min. However, the signal intensity was reduced ten-fold and would require higher specific labeling of the primer or longer exposure time to compensate. While the product distribution at 200 min followed an exponential decay, the calculated processivity now slightly under-estimated the processivity (P = 0.88, R_1/2_ = 5.7 repeats). These systematic errors can be accounted for as follows. In the case of primer depletion, products have a second chance to rebind and become extended and hence increase the apparent processivity. In the case of single hit conditions, the early repeat products have a head-start in accumulation compared with later repeats and hence are overrepresented in the processivity calculation, leading to a decrease in the apparent processivity. For a fixed number of bands, *n*, under analysis, longer incubation times would reduce this error, but then the chances of multiple hits increase. In conclusion, the use of an unlabeled primer chase is the best solution as it avoids these ambiguities.

The ratio of primer to telomerase of 10 used in the simulation in Figure 4a is higher than many experimental protocols but is close to that used by Jansson et al. [12]. While this ratio results in departure from single hit conditions, the burst in primer binding provides a sensitive measure of telomerase active site concentration which is generally not well-defined by the protein concentration.

### 2.5. Analysis of Nucleotide Addition Processivity (NAP)

Figure 2 was also used to model the intervening NA-transitions with the appropriate assignment of the microscopic processivity values at each step. In primer extension assays (Figure 1b), the NA-bands are typically about 10 to 20% of the intensity of the adjacent RA-bands for 10 μM dNTP concentrations [12]. This indicates that processivity associated with NAP is significantly higher for RAP. In the following simulations, we assigned a microscopic processivity = 0.99 for all NA-transitions in order to determine any systematic deviations of the extracted processivity values. To aid comparison with the RAP-only simulations above, the processivity of the G6 to G1-transition was set to 0.9464, so that the overall RAP = 0.9, when analyzed as a single step. This value was derived from the cumulative relationship of the processivity: (i.e., 0.99^5^ × 0.9464 = 0.90). Note that a RA-band, as defined here, requires the preceding NA-step (the incorporation of G6) in order for it to be visible on a gel, while the stalling due to the translocation steps slows the incorporation of the next G1. Figure 5a shows the expected product distribution when the reaction approaches equilibrium, calculated for 1000 steps with P^NAP^ = 0.99 and P^RAP^ = 0.9464. Fitting the intensity of the RA-bands only to an exponential function gave a decay constant of 0.1 repeat^−1^, corresponding to an overall processivity of 0.90. Plotting the same data according to von Hippel [4] yielded P^NAP^ = 0.99 and P^RAP^ = 0.945 from the intercept values, while the slope gave the cumulative processivity of the combined NA- and RA-events = 0.90 (Figure 5b). When intensities corresponding to the NA- and RA-products were plotted according to Latrick and Cech [7], the transitions between the NA- and RA-events were greatly smoothed. Small discontinuities were observed at each RA-step, but the overall trend was linear with a gradient of −0.1 repeat^−1^ (Figure 5c). The same gradient was observed when the intensities for the RA-bands alone were plotted and yielded a processivity of 0.90. A Peng plot (Figure 5d) of the NA- and RA-bands yielded the input microscopic processivities of p^NAP^ = P^NAP^ = 0.99 and p^RAP^ = P^RAP^ = 0.9464. When the RA-bands were analyzed as a single transition, the observed p = P = 0.90, regardless of whether the intensity was based on that of the RA-bands alone, or that summed with the adjacent NA-bands to retain conservation of mass.

The increased processivity of NAP relative to the RAP could arise from changes in *k_f_* and/or *k_d_*. Three models were considered (i) *k_d_* remained similar and the reduced band intensities for NAP reflects the shorter transition time to the next intermediate because of an increased *k_f_* (ii) *k_f_* remains similar and the reduced band intensity for the NA-bands arises from a smaller *k_d_* (iii) both *k_f_* and *k_d_* are higher for the NA-transitions, but *k_f_* is most affected to give a higher processivity. These scenarios showed different kinetics but, with the appropriate selection of rate constants, the end-point product distributions can be identical. Hence, data obtained just from near end-point assays cannot distinguish these mechanisms.

Rate constants for the simulations were selected such that the processivity at each step matched those of the equilibrium models discussed above (i.e., P^NAP^ = 0.99 and P^RAP^ = 0.9464) while the overall transit time between RA-bands was comparable to that of the RAP-only model (1/0.18 = 5.5 min) of Figure 3. The resultant schemes are shown in Figure 6. Simulations of these mechanisms showed clear differences (Figure 7). When *k_d_* was similar for the NA- and RA-transitions, the NA-product concentrations remained at a near constant ratio (15 to 20%) compared with the neighboring RA-products throughout the time course (Figure 7a). When *k_f_* was similar for the NA- and RA-transitions, the NA-product concentrations showed transient values which were comparable to the peak RA-products (Figure 7b). When *k_f_* and *k_d_* were larger for the NA-transitions, the NA-product concentrations reached their near-end point values without any transient phase, apart from the small peaks for the NA-transitions before the first RA-transition (Figure 7c). The time courses reported by Jansson et al. [12] and Figure 1b most closely followed the model of Figure 7a, indicative of the slowing of the forward transitions associated with the RA-transition, while the dissociation rate constants were less affected. The latter observation indicates that the anchor site is effective at retaining the DNA on the telomerase during the translocation steps.

A challenge for the analysis of experimental data is that the NA-bands are only well-resolved between early RA-bands (typically up to about the 5th or 6th repeat) and, beyond the 10th repeat, quantification of NA-bands is difficult (Figure 1b and Appendix A). The low intensities and poor resolution of the NA-bands lead to large relative errors in their estimated contribution, particularly for high molecular weight products. In the analysis of experimental data, the NA-band intensities can be ignored, pooled with adjacent RA-bands, pooled with adjacent NA-bands or subject to full analysis. Analysis of simulated data provides an assessment of the potential errors introduced by these approaches. For this purpose, the scheme of Figure 6b, with NA-events k_f_ = 1.98 min^−1^, RA-events k_f_ = 0.353 min^−1^ and k_d_ = 0.02 min^−1^, was subject to further analysis. Following the simulation as shown in Figure 7a (determined with a numerical time increment of 0.00002 min), the data set was reduced by taking concentration values at 5 min intervals from 0 to 40 min following the chase, to make the set comparable to experimental sampling times. With this limited data set, DynaFit returned accurate rate constant values for both NA- and RA-steps for the first three RA-steps and the intervening NA-steps (deviation from input values was <1%: Appendix A). To simulate the experimental noise and distortion of NA-band intensity by adjacent dominant RA-bands, a random factor was added to or subtracted from each band concentration with a value up to 10% of the summed concentration of the band with its neighbors (Appendix A). In this way an error in defining the boundary between a RA-band and the adjacent NA-band would distort the latter to a greater relative extent, as might occur in the experimental analysis of gel intensities. In this case, the deviations in returned rate constants for NA-steps k_f_ and k_d_ were <40% of the input values while for RA-steps *k_f_* and *k_d_* they were <20% (Appendix A). Note that the estimates of the rate constants for the first 3 NA-steps prior to the first RA-transition had a large error because these steps were largely complete by the first time point in the simulation (5 min after mixing when the chase was initiated). These simulations indicate that rate constants can be determined for the individual NA-transitions provided the NA-band intensities can be estimated with high accuracy, but this condition is unlikely to be met with experimental data beyond the first few repeats (cf. Figure 1b and Appendix A).

For later repeats, where the NA-bands are poorly resolved, the intensity of the NA-bands between each pair of RA-bands can be summed and the NA-transitions modelled as a single elementary step. This approximation leads to a systematic deviation of the fit whose magnitude was ascertained by simulation. In the example in Appendix A, the five sequential NA-transitions, each with a rate constant of 1.98 min^−1^, are reduced to a single step with an apparent rate constant ≈ 0.39 min^−1^ (Appendix A). When simulated data were analyzed using DynaFit, the returned values of *k_f_* and *k_d_* for the first 7 RA-transitions deviate from the input values (0.353 min^−1^ and 0.02 min^−1^ respectively) by <10% and <20% respectively (Appendix A). These deviations are of the same order as the likely random experimental error and similar to the reported standard errors in the fit. However, the deviations are systematic and increase with the RA repeat number. They also depend on the precise time intervals over which the reaction is sampled. Fits to early RA-transitions place more emphasis on the end-point concentrations compared with later transitions, when sampled over the same intervals.

A further simplification can be made by pooling the NA-band intensities with the adjacent RA-band, which allows analysis of a large number of RA-steps. Again, the effect of this approximation was ascertained by analysis of simulated data (Appendix A). In the example analyzed, the NA- and RA-transitions have constant input rate constants and consequently it makes little difference (apart from the first transition) whether the NA-intensities are summed with the preceding or trailing RA-intensities or split between the two. The latter is the most practical method for high repeat numbers when only RA-bands can be resolved and was used in the analysis here. As discussed above, 5 consecutive NA-transitions with *k_f_* = 1.98 min^−1^ coupled with 1 RA-transition with *k_f_* = 0.353 min^−1^ approximates to a single transition with *k_f_* ≈ 0.18 min^−1^ (Figure 6a). The fitted values of *k_f_* and *k_d_* for the 2nd to 7th RA-transition deviate by <25% and <40% respectively from the expected input values (Appendix A). Monte-Carlo analysis shows that *k_f_* and *k_d_* were covariant so that the deviations of the calculated microscopic processivity values from the input value (=0.9) is substantially less (<3%). Again, the estimated rate constants show systematic deviations from the true values for the same reasons as discussed above. The reported standard errors in the DynaFit estimates encompass the absolute deviations of the output which arise from lumping the NA-products with the adjacent RA-products. Ignoring the NA-products completely (Appendix A) is the poorest approximation (Appendix A), as well as presenting a practical challenge for high repeat numbers where the resolution of the NA-bands from the RA-bands is poor. Nevertheless, the extracted microscopic processivity values are still within 6% of the expected value.

For the analysis of experimental data, the above simulations suggest the following strategies: (i) When NA-bands are sufficiently resolved, it is best to include them in the analysis. Even if the estimated rate constants for the NA-transitions have a large error (as indicated by the standard error of the fit or confidence interval from Monte Carlo analysis), the rate constants for the RA-transitions (G6 to G1) are subject to minimal systematic error; (ii) If analysis extends into regions where NA-bands are poorly resolved, they are best treated as a single species (i.e., lump the 5 NA-band intensities into one); (iii) At high repeat numbers where NA-bands cannot be distinguished from the RA-bands, then the estimated RA-band intensities necessarily include the unresolved NA-intensities. When analyzing data using a simple processive model of RA-transitions (Figure 6a), the intensities of the partially-resolved NA-bands at lower repeat numbers should be added to the adjacent RA-intensities rather than ignored (cf. Appendix A). The latter results in the failure to conserve mass and introduces the largest systematic error.

## 3. Discussion

The analysis of simulated time courses for telomerase activity helps in experimental design and identifies the potential systematic errors in various fitting procedures to determine the processivity characteristics. These systematic errors can be of the same order or greater than the random error in data from well executed experiments. For screening purposes, where a single macroscopic processivity value is required for comparing different conditions or preparations, a processivity index may suffice. Rather than picking on the fraction of product which exceeds an arbitrary nucleotide length or repeat number [15,24], a model-independent R_1/2_ value for labelled primer assays can be determined directly by determining the band number that represents 50% of the total product intensity (e.g., cumulative pixel intensity determined from a gel scan as in Appendix A). Alternative analytical methods that assume the product distribution follows an exponential function are subject to biased error weighting, particularly when the data are linearized by plotting on a logarithmic scale [21,22]. Furthermore, for methods that involve normalization to the total product intensity [5,7], it is important to include all products and not just that estimated from the sum of the resolved bands (Figure 2c,d and Appendix A). The systematic error associated with an underestimate of the total product is evident in the downward curvature of Latrick-Cech plot with increasing repeat number seen in some experimental records [7,25]. This curvature, which may involve a 2-fold change in slope, leads to an uncertainty in the mean processivity value which is much greater than the precision of the individual data points.

In general, the band intensities on a gel represent the sum of free and bound product present at the time of quenching. While physical methods may be used to separate these products prior to electrophoresis [23], it is better to analyze just those bands which have come to their near equilibrium values. In a pulse-chase experiment, the labelled products then represent just the free DNA. In practice, this means using an incubation time such that the resolved bands no longer show any change in intensity with time within the limit of detection. Without a cold chase with excess primer, distribution of products may appear to follow an exponential-like distribution, but the distribution profile for the total product is distorted due to the time delay in forming the higher molecular weight species. Furthermore, the labelled primer will become depleted and may allow products to rebind and undergo further extension, so complicating any estimate of processivity. While such data may be analyzed by fitting to a kinetic scheme and assuming the DNA products have similar binding kinetics as the primer, such an assumption is best avoided. A non-chase experiment with a relatively high enzyme:primer ratio (e.g., 1:10) is, however, a useful assay in that the amplitude of the product burst kinetics provides a measure of the active telomerase concentration [12].

For detailed mechanistic analyses, microscopic processivity values at each step in the reaction are more informative than a single “average” macroscopic value. Here the analysis of Peng at al. [5] is useful, given the experimental caveats discussed above. Full kinetic analysis of primer extension assays provides more information [12] and should lead to end-point values that are compatible with the Peng et al. [5] estimates. In particular, such kinetic analysis reveals that the more intense bands associated with RAP arise from a slowing of the forward elongation reaction associated with the translocation process, rather than an increased dissociation probability. The later might seem a reasonable possibility on structural grounds in that the DNA-RNA duplex must transiently dissociate (or at least loop out [11]) to allow repositioning of the DNA on the RNA template, ready for the next round of elongation. The observation that the dissociation rate constant is little affected, argues that the anchor site(s) which keeps the DNA tethered to the telomerase, plays a dominant role in the total DNA-telomerase interaction relative to the DNA-RNA base pairing itself [9,10]. Nevertheless, this anchor site must be dynamic to accommodate the growing DNA strand and means that dissociation can occur with a similar probability at any step of the reaction, thus contributing to the moderate processivity values compared with many DNA polymerases. Interestingly, recent structural data for ciliate and human telomerases reveals a wedge in the active site that separates the RNA-DNA duplex during synthesis, such that only about 4 to 6 nucleotides may form base pairs with the RNA template throughout the nucleotide addition cycles [9,10,26]. Such a mechanism would aid the energetics of translocation, because it spreads the energetic cost of strand dissociation over multiple steps. This possibility was recognized in the early research on telomerase [6].

Although we treat the forward elongation step as a single elementary step in our analysis, it must comprise multiple steps, including: dNTP binding, reaction of the dNTP with the DNA primer, release of pyrophosphate product and movement of the DNA-RNA hybrid by one base pair to bring the next free template nucleotide into the active site (Figure 1a). Non-cognate dNTP may also bind weakly and occasionally become incorporated. In addition, at the end of a repeat synthesis cycle, there are additional steps which involve dissociation of the DNA strand and movement of the RNA and DNA strands with respect to the active site (or movement of the duplex followed by strand dissociation) and re-engagement of these strands. Previous single-molecule studies [27] suggested that the DNA strand translocation step is relatively rapid compared with the overall transition time associated with RAP. On the other hand, at the relatively low concentrations of dNTPs used in most assays, nucleotide binding is likely to be a significant rate-contributing step. Indeed, Chen at al. [23] considered that the *K_m_* for the incorporation of the dGTP at the C1 template position is relatively high (~120 to 160 µM) compared with other sites on the template (5 to 30 µM). Therefore, at low dNTP concentrations, this would have the effect of selectively slowing the incorporation event for G1 and thus account, at least in part, to the increased intensity of the RA-bands compared with the NA-bands. If this were the sole difference between the NA- and RA-bands then, at increasing [dNTP], the NA-band intensity should approach that of the RA-bands. In practice, while a dependence in this direction is seen in some conditions [23,28], it does not seem to contribute in all cases [14]. It is likely that a first-order transition (i.e., zero order with respect to dGTP), involved with the overall reactions associated with telomerase translocation, also contributes. In this regard it of interest to note that recent zero mode waveguide measurements, which follow nucleotide incorporation at the single-molecule level, reveal pauses prior to as well as after the binding of G1 (and G2?) positions which delay incorporation and slow down forward progression [29]. It should also be noted that the term “rate-contributing (or rate-limiting) step” is over simplistic and can lead to misinterpretation. A high *K_m_* for dGTP for G1 incorporation might imply a weak affinity for the nucleotide. However, such an observation could be accounted for by inherently similar dNTP binding characteristics as the other sites, but that telomerase at this point in the cycle is in rapid equilibrium between competent and non-competent states, with the latter being dominant (in effect, a conformational selection model). Such non-competent states could arise during the translocation when either or both the template RNA and DNA are not in their correct sites for further elongation. Thus, an unfavorable rapid equilibrium step in combination with a moderately fast nucleotide binding step, gives rise to an effectively slow binding process (and high apparent *K_m_*) and it is inappropriate to identify either of the individual steps as “rate limiting” [21]. While Parks and Stone [27] found that DNA movements during translocation were rapid, they estimated the equilibrium position lay in favor of the forward reaction. However, this movement is only one part of the translocation process and the equilibria of other steps could lie in the unfavorable direction [11].

The systematic deviations arising from truncation of experimental records with respect to assay time and electrophoretic mobility, as reported in Figure 2 and Figure 3, affect the higher molecular weight products. In practice, experimental data indicate that the processivity of the first one or two repeats is lower than subsequent repeats [12], and these bands are often omitted for the estimate of the median processivity of the remaining higher molecular weight products [7]. This observation likely arises from the increased dissociation rate constant for the early repeat RA-products [12], suggesting that the shorter DNA products make less extensive interactions with the anchor sites on the telomerase. Other systematic errors, such as a less efficient precipitation of low molecular weight products during sample preparation [30], would have the effect on increasing the apparent processivity of the low molecular weight species. We found that the estimates of the rate constants for elongation from gel-based assays showed good agreement with single-molecule assays for the 2nd to 5th repeat [29], suggesting that systematic errors introduced by the gel-based procedure are not dominant factors in this size range.

In summary, we have used kinetic simulations to critically examine the strengths and limitations of established analytical procedures for determining the processivity of telomerases. This exercise also helps in experimental design in order to maximize the information content and minimize ambiguities. While recent structural data have greatly contributed to the understanding of telomerase mechanism, novel kinetic assays are required developed to interrogate individual steps in the cycle.

## 4. Materials and Methods

### 4.1. Calculation of End-Point Distributions

The calculation of the end-point product distribution (i.e., that at infinite time) for Figure 2 for a given macroscopic processivity, or set of microscopic processivities, is easily accomplished using a spreadsheet. Consider Figure 2 where the microscopic processivity is the same for all steps and equals 0.9. At each step in the reaction there is a 90% chance of going forward to the next elongated product and a 10% chance of dissociation. Under conditions where product dissociation is effectively irreversible (e.g., in a pulse-chase assay), the relative concentrations of the dissociated products will follow a single exponential distribution, as evident from the calculation shown in Table 1.

In the experimental situation, a finite incubation period must be chosen. In practice, this time is determined from that period where the change in product distribution is no longer experimentally measurable. The experimenter must determine the cut-off point above which bands are not sufficiently resolved from each other and focus on analysis of bands below this cut-off. Nevertheless, when the data are normalized to the sum of product band intensities [5,7], it is important to include the unresolved bands in the total intensity calculation (Appendix A).

### 4.2. Simulation of Kinetic Schemes

Time courses for the model shown in Figure 2 were simulated using numerical integration, with concentrations and rate constants selected to be close to those characterized experimentally [12]. While several kinetic simulation packages are available [21], we used Berkeley Madonna (https://berkeley-madonna.myshopify.com/ accessed on 7 December 2021) for simulations. An example script is provided in the Appendix A. As a test for systematic errors introduced by the various analysis procedures, Figure 2 was modelled with the same values for the rate constants *k_f_* = 0.18 min^−1^ and *k_d_* = 0.02 min^−1^ for each RA-step, corresponding to a processivity = 0.9. Pulse-chase simulations, involving the addition of excess cold primer, were modelled using the Pulse function. The resultant simulations were then subject to analysis as below.

### 4.3. Analysis of Simulated Kinetic Data

Simulated data were analyzed to determine how closely the extracted rate constants matched the input rate constants. To avoid potential circular logic, we used DynaFit (http://www.biokin.com/dynafit/ accessed on 7 December 2021; [16,17]) to fit the simulated data and extract rate constants by non-linear least squares fitting. This software uses LSODE (https://computing.llnl.gov/sites/default/files/ODEPACK_pub2_u113855.pdf accessed on 7 December 2021) as the default algorithm for numerical integration, as opposed to Runge-Kutta4 used by Berkeley Madonna. Furthermore, data were simulated using a step size Δt ≤ 0.02 min, while time courses subject to analysis were limited to data selected at ≥5 min intervals, to match typical experimental times. In some cases, random noise was added to the simulated data to test the robustness of fitting. An example of a DynaFit script is provided in the Appendix A. Analysis of schemes having 10 steps took several minutes, while estimating the confidence intervals using 1000 Monte Carlo interactions took several hours on a standard laptop computer. For models containing more steps, the simulations can be broken down into two phases, where the fits to the first 10 steps are used as fixed parameters in the fitting of the second set. This process is valid when the nucleotide addition steps (*k_fi_*) are essentially irreversible, so that the elongation steps become decoupled and there is no influence of later products on the initial ones.

## Data Availability

The data presented in this study are available on request from the corresponding authors.

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
