# Peer review of "The Processivity of Telomerase: Insights from Kinetic Simulations and Analyses"

_molecules, 2021, doi:10.3390/molecules26247532_

Round 1

Reviewer 1 Report

The authors described how they can make more reliable quantification of the processivity of a telomerase by the corrections proposed in this study. They propose a method to evaluate the amounts of DNA products of various lengths from the a autoradiogram(s), making several corrections. Therefore, the value of this study depends on how significant these corrections are.

However, it is impossible to judge the significance from the present manuscript due to the lack of the essential descriptions on the wet process and and essential control experiments to deny possible artifacts.  I hope author to understand the importance of describing “wet factors” and how modest we experimentalists must be in interpreting our own data, because they can be heavily affected by unknown and unaware artifacts. Therefore, this manuscript cannot be accepted in this form.

The methods for obtaining the data shown in Fig. 1b are not described. Thus, the following discussion is under the assumption that the methods is the same as described in Ref. 12 or the data are transferred from the reference, the author of which involves the authors of this manuscript (they should make clear according to the indications to the author). The largest difficulty is that the catalytic products were phenol-chloroform extracted and then ethanol precipitated. Although this is a standard preparation for electrophoresis of nucleic acids, to remove salts and concentrating the nucleic acids to get sharp bands. However, these treatments are known to have significant loss of the samples and the yield is length-dependent. For quantifying purpose, these precess should be avoided. For example, ethanol precipitation shows a less yield for a DNA shorter than 20 base. Therefore, they may underestimate the amount of short products ~10 base long, leading overestimation of the processivity of the step from 10 base to 20 base. The yield depends on not only the length but also the ethanol concentration at precipitation as well as wash, initial salt concentrations, existence of co-precipitants, and centrifugal force. In my experience, 30~40 % less yield than that for long nucleotide can easily happen by the extract and the precipitation for 10 base oligo-nucleotide. At least, they need to describe the evidence that the possible underestimation due to the treatments is much smaller than the theoretical corrections they propose. I am afraid that the control experiments are difficult because the experiments are typically made in a volume of 15 ul, too small to reproduce the same concentrations of the components. This is the reason why most experiments in molecular biology has only 1.5 digid of accuracy. In addition to the poor description on the wet process, the discussion on the obtained value of processivity is poor and the purpose of this study is not clearly contrasted.

Author Response

Reviewer #1

  1. The authors described how they can make more reliable quantification of the processivity of a telomerase by the corrections proposed in this study. They propose a method to evaluate the amounts of DNA products of various lengths from the a autoradiogram(s), making several corrections. Therefore, the value of this study depends on how significant these corrections are.

We have addressed this potential problem as outlined in the comments to the Editor above.

  1. However, it is impossible to judge the significance from the present manuscript due to the lack of the essential descriptions on the wet process and and essential control experiments to deny possible artifacts. I hope author to understand the importance of describing “wet factors” and how modest we experimentalists must be in interpreting our own data, because they can be heavily affected by unknown and unaware artifacts. Therefore, this manuscript cannot be accepted in this form.

The methods for obtaining the data shown in Fig. 1b are not described. Thus, the following discussion is under the assumption that the methods is the same as described in Ref. 12 or the data are transferred from the reference, the author of which involves the authors of this manuscript (they should make clear according to the indications to the author).

The legend to Figure 1b is now clarified. The gel was run by JH using essentially the same protocol as described in reference 12 (where the gels were run by a previous colleague. L. Jannson).

  1. The largest difficulty is that the catalytic products were phenol-chloroform extracted and then ethanol precipitated. Although this is a standard preparation for electrophoresis of nucleic acids, to remove salts and concentrating the nucleic acids to get sharp bands. However, these treatments are known to have significant loss of the samples and the yield is length-dependent. For quantifying purpose, these precess should be avoided. For example, ethanol precipitation shows a less yield for a DNA shorter than 20 base. Therefore, they may underestimate the amount of short products ~10 base long, leading overestimation of the processivity of the step from 10 base to 20 base.

We thank the referee for pointing out this potential systematic error in experimental data. As stated in the comments to the editor, the ssDNA bands analyzed in experiments represent an extension of an existing primer and exceeded 20 base pairs.

  1. The yield depends on not only the length but also the ethanol concentration at precipitation as well as wash, initial salt concentrations, existence of co-precipitants, and centrifugal force. In my experience, 30~40 % less yield than that for long nucleotide can easily happen by the extract and the precipitation for 10 base oligo-nucleotide. At least, they need to describe the evidence that the possible underestimation due to the treatments is much smaller than the theoretical corrections they propose.

There was no indication of a systematic change in the total band intensity as a function of time (and hence molecular weight distribution over the range 20 to 160 bases), following a cold chase, within the 10% variation which could arise from errors in the extraction and gel loading. We agree there are limitations within the experimental data, particularly in estimating accurate intensities associated with the weak nucleotide addition bands, which prevents full benefit from a detailed kinetic analysis. However, the systematic deviations arising from errors in the analytical procedures, such as normalizing to an incomplete product sum are comparable to or exceed experimental “noise”.

  1. I am afraid that the control experiments are difficult because the experiments are typically made in a volume of 15 ul, too small to reproduce the same concentrations of the components. This is the reason why most experiments in molecular biology has only 1.5 digi[ts] of accuracy. In addition to the poor description on the wet process, the discussion on the obtained value of processivity is poor.

We discussed that the processivity associated with the repeat addition bands is lower than the intervening nucleotide addition bands because of a “stalling” of the forward reaction as opposed to an increase in the dissociation rate constant for intermediates involved in the translocation process. This is of interest in the context of recent structural data concerning the nature of the anchor site(s). As far as we know, this conclusion has not been drawn by previous workers and the purpose of this study is not clearly contrasted.

The purpose of this study was stated in several places, for example:

“Microscopic processivity values were determined from kinetic analysis of time courses using non-linear least squares regression (DynaFit [16,17]) for the model shown in Equation (2). Here, we present the details behind this procedure and test the DynaFit program using simulated data to check for factors that can affect the analysis. In addition, these simulated data allowed a quantitative assessment of systematic errors that can arise in other established methods for determining processivity values.”

“While a single macroscopic processivity value or index is useful for comparative purposes for determining the effect of conditions, regulators or mutations on processivity, microscopic processivity values are required for a mechanistic understanding of the effect of local sequence and structure on the catalytic activity of telomerases.”

In particular, the current manuscript builds on our previously published work [12] where, due to space restrictions, the analysis procedures we employed were briefly summarized in the supplementary text. It was also inappropriate to review other established procedures for analyzing processivity in reference [12].

Reviewer 2 Report

Bagshaw and co-workers show a very nice work on the processivity of telomerase. Specifically, they tentitavely to solve how product concentrations increase systematic errors in the processivity estimate. With primer extension assays and simulation of kinetics schemes, they show that higher product concentrations associated with repeat addition processivity arise from a stalling of nucleotide incorporation rather than an increased rate constant for the dissociation of DNA. In general, I think the manuscript is neatly written and adds new knowledge to the field. I recommend its publication in Molecules after addressing the following minor issues.

  1. The figure caption of Figure 1b is not provided.
  2. Figures 2~5 should better be revised. For instance, the size of tick label of Figure 2 is too small.

Author Response

Reviewer #2

Bagshaw and co-workers show a very nice work on the processivity of telomerase. Specifically, they tentitavely to solve how product concentrations increase systematic errors in the processivity estimate. With primer extension assays and simulation of kinetics schemes, they show that higher product concentrations associated with repeat addition processivity arise from a stalling of nucleotide incorporation rather than an increased rate constant for the dissociation of DNA. In general, I think the manuscript is neatly written and adds new knowledge to the field. I recommend its publication in Molecules after addressing the following minor issues.

  1. The figure caption of Figure 1b is not provided.

Panel (b) was described in the original manuscript: “(b) Example of a primer extension assay ….”. We have now made the panel letters throughout in the legends bold which makes it easier to associate the text with each figure panel, and clarified the relation of Figure 1b to reference [12]

  1. Figures 2~5 should better be revised. For instance, the size of tick label of Figure 2 is too small.

The axis and tick marks have been thickened to make them stand out. Please note the line thickness in the figures embedded within the text is affected by the exact placement of the figure in the panel and should be enlarged to > 100% to see the true relative thickness. Top copy figures are provided as separate documents.

Reviewer 3 Report

This work deals with the kinetic simulations and analyses of an enzymatic reaction catalyzed by telomerase. The topic is very interesting and the results are presented properly as graphics. However, this manuscript needs minor revisions.

General Comments
1. On line 43 it is presented Scheme 2, but it is not found in the manuscript. Line 45 starts with Equation (2), but it is not any equation. Equation (2) appears also in line 168. Please correct the problem.

2. Please provide an explanation about Figure 1 b.

3. Data from Apendix A should be part of Supplementary Material.

Author Response

Reviewer #3

 This work deals with the kinetic simulations and analyses of an enzymatic reaction catalyzed by telomerase. The topic is very interesting and the results are presented properly as graphics. However, this manuscript needs minor revisions.

General Comments

  1. On line 43 it is presented Scheme 2, but it is not found in the manuscript. Line 45 starts with Equation (2), but it is not any equation. Equation (2) appears also in line 168. Please correct the problem.

Equation (2) has been replaced by Scheme (2) throughout, as has Equation (1) with Scheme (1).

  1. Please provide an explanation about Figure 1 b.

Figure 1b is included to show the characteristic nucleotide-addition (NA-) and repeat-addition (RA-) bands. The experimental details follow the same protocol as in reference [12], as is stated more explicitly.

  1. Data from Apendix A should be part of Supplementary Material.

Content of the Appendices has been moved to Supplementary Material.

Round 2

Reviewer 1 Report

As an experiment expert, I feel odd to this manuscript. For example describing the results in 3 digits, which is calculated from the experiment where pipetting accuracy is worse than 1%. More importantly, the reproducibility among the independent runs of the same experiment are not described at all. This manuscript is very finely described but the analysis is made for the results obtained a particular single run of experiment (only ambiguously specified). If the authors want to appeal the usefulness of their conclusion, they must show the reproducibility for generality.

The response to my question on ethanol precipitation was fully answered, but the response to phenol extraction was not appropriate. The total count does not mean much because the loading amount can be adjusted. The authors seem to lack the understanding of the quantitative effect of the extraction. The soluble substance tend to remain in the top aqueous layer, and proteins are denatured to be aggregated and tends to remain near the interface. However, the separation is imperfect and insoluble aggregates retaining nucleic acids as well as soluble nucleic acids significantly exist in both layers as emulsion. Therefore, phenol extraction generally spoils reproducibility although more clear bands are obtained by the treatment. The background of the bands are not due to the diffusion of each band component but due to slow solubilization of nucleic acids from aggregates retained at the bottom of the gel slot. The radioactivity is left over at the bottom indicates the formation of the aggregate, but it is not shown in this manuscript. Usually the aggregate formation is not reproducible and difference among autoradiograms is not small. 

However fine, description is not the purpose of science. 
